# Emerging Role of Phospholipids and Lysophospholipids for Improving Brain Docosahexaenoic Acid as Potential Preventive and Therapeutic Strategies for Neurological Diseases

**DOI:** 10.3390/ijms23073969

**Published:** 2022-04-02

**Authors:** Mayssa Hachem, Houda Nacir

**Affiliations:** Department of Chemistry, Khalifa University, Abu Dhabi P.O. Box 127788, United Arab Emirates; houda.nacir@ku.ac.ae

**Keywords:** docosahexaenoic acid, phospholipids, lysophospholipids, cerebral accretion, neurodegenerative diseases

## Abstract

Docosahexaenoic acid (DHA, 22:6n-3) is an omega-3 polyunsaturated fatty acid (PUFA) essential for neural development, learning, and vision. Although DHA can be provided to humans through nutrition and synthesized in vivo from its precursor alpha-linolenic acid (ALA, 18:3n-3), deficiencies in cerebral DHA level were associated with neurodegenerative diseases including Parkinson’s and Alzheimer’s diseases. The aim of this review was to develop a complete understanding of previous and current approaches and suggest future approaches to target the brain with DHA in different lipids’ forms for potential prevention and treatment of neurodegenerative diseases. Since glycerophospholipids (GPs) play a crucial role in DHA transport to the brain, we explored their biosynthesis and remodeling pathways with a focus on cerebral PUFA remodeling. Following this, we discussed the brain content and biological properties of phospholipids (PLs) and Lyso-PLs with omega-3 PUFA focusing on DHA’s beneficial effects in healthy conditions and brain disorders. We emphasized the cerebral accretion of DHA when esterified at *sn-2* position of PLs and Lyso-PLs. Finally, we highlighted the importance of DHA-rich Lyso-PLs’ development for pharmaceutical applications since most commercially available DHA formulations are in the form of PLs or triglycerides, which are not the preferred transporter of DHA to the brain.

## 1. Introduction

Docosahexaenoic acid (DHA, 22:6n-3) is the main polyunsaturated fatty acid (PUFA) in brain tissues essential for normal brain development and function. DHA bioconversion from alpha-linolenic acid (ALA, 18:3n-3) is limited in mammals (less than 1% in human [1]). This endogenous supply of DHA does not compensate for its cerebral deficiencies in patients suffering from neurodegenerative diseases. An exogenous resource of DHA is required to compensate for these deficiencies. Therefore, new therapeutic approaches to target the brain with DHA are needed, taking into consideration the difficulties associated with delivering materials across the highly restrictive blood–brain barrier (BBB) [2].

For many decades, researchers have highlighted the importance of brain targeting with DHA for prevention and potential treatment of neurodegenerative diseases.

The cerebral bioavailability of the physiological form of DHA including non-esterified DHA, DHA esterified in lysophospholipids (Lyso-PLs), phospholipids (PLs), and triglycerides (TGs) were investigated. In addition, researchers suggested some structured phospholipids as a vehicle of DHA to the brain [2,3,4,5].

In this context, researchers showed through in vitro and in vivo studies that 1-lyso,2-docosahexaenoyl,glycerophosphocholine LysoPC-DHA was a favored functional carrier of DHA to the brain when compared to non-esterified DHA [6,7]. This preference was only observed in the brain but not in the liver, kidney, or heart, which even prefer the free form of DHA [7]. Additionally, researchers considered the cerebral accretion of a structured phosphatidylcholine to mimic Lyso-PC-DHA, named AceDoPC^®^ (1-acetyl, 2-docosahexaenoyl-glycerophosphocholine), which can be considered a stabilized physiological Lyso-PC-DHA [8]. Moreover, the bioavailability of DHA esterified at *sn-2* position of different PLs including phosphatidylserine (PS-DHA), phosphatidylcholine (PC-DHA), and phosphatidylethanolamine (PE-DHA) were considered suggesting that PS and PC were the privileged carriers of DHA to the brain in comparison to PE [9]. Thus, diverse mechanisms could explain these differences in cerebral bioavailability of different DHA lipid carriers.

Researchers studied the cerebral accretion of different PL-DHA and Lyso-PC-DHA; however, to our knowledge, the brain bioavailability of different forms of Lyso-PL-DHA in comparison to free DHA or other lipids’ forms of esterified DHA have not yet been explored.

The present review focused on the emerging role of several lipids carrying DHA to the brain for potential prevention and treatment of neurodegenerative diseases. Since glycerophospholipids (GPs) are essential constituents of the cell membrane and participate in DHA transport, to better understand their biochemistry, we first introduced de novo GPs biosynthesis including phosphatidic acid (PA), phosphatidylcholine (PC), phosphatidylethanolamine (PE), phosphatidylserine (PS), phosphatidylglycerol (PG) and phosphatidylinositol (PI) biosynthesis. In addition to de novo synthesis, alternative pathways responsible for fatty acids’ remodeling in PLs such as de-acylation, re-acylation, and transacylation pathways were covered in this review. Following this, we discussed the main PLs and their fatty acids composition found in the brain. Since DHA can also be esterified in cerebral Lyso-PLs, properties and biological functions of several Lyso-PLs including lysophosphatidylcholine (Lyso-PC), lysophosphatidylethanolamine (Lyso-PE), lysophosphatidylserine (Lyso-PS) and lysophosphatidic acid (Lyso-PA) were highlighted. After discussing DHA’s benefits in healthy conditions and cerebral disorders, we focused on DHA’s transport to the brain and effects when esterified at *sn-2* position of several GPs including PLs, structured PLs, Lyso-PLs, and TGs. We explained the chemical, biochemical, and nutritional properties of the *sn-2* position. Finally, we focused on the need to develop and compare the cerebral accretion of Lyso-PC, Lyso-PE, Lyso-PS, Lyso-PI, Lyso-PG, and Lyso-PA in comparison to the non-esterified form of DHA.

## 2. Glycerophospholipids

### 2.1. Biosynthesis of Glycerophospholipids

Glycerophospholipids (GPs) are vital elements of cell membranes since they represent 80% of total membrane lipids. These lipids contain fatty acids (FAs) at *sn-1* and *sn-2* positions of a glycerol backbone and a phosphate moiety at *sn-3* position on which a polar head is esterified. GPs are amphiphilic molecules with a phosphate group and polar head, which constitute the hydrophilic part, while acyl chains and glycerol form the hydrophobic moiety. Regarding FA composition, *sn-1* position is often occupied by a saturated fatty acid whereas an unsaturated (mono or polyunsaturated) fatty acid is found at *sn-2* position. Additionally, FA composition of GPs influences the membrane’s fluidity; the shorter and more unsaturated the carbon chains of FAs are, the more fluid the membrane.

In each tissue, cell membranes contain distinct composition of different GPs.

Via the de novo route, different GPs with diverse polar heads at *sn-3* position of glycerol, such as phosphatidic acid (PA), phosphatidylcholine (PC), phosphatidylethanolamine (PE), phosphatidylserine (PS), phosphatidylglycerol (PG), and phosphatidylinositol (PI) are generated [10]. Figure 1 illustrates the structure of common GPs.

Kennedy and Weiss were the first to describe GPs’ biosynthesis or de novo synthesis in 1956. De novo synthesis starts with sn-glycerol-3-phosphate synthesis through dihydroxyacetone phosphate’s reduction or in a minority by glycerol’s phosphorylation. In this section, we discussed de novo synthesis of different PLs including PA, PC, PE, PS, and PG along with the main cerebral metabolic pathways involved.

Phosphatidic acid (PA) is the common intermediate in all PLs’ synthesis. Lysophosphatidic acid is first formed from *sn*-glycerol-3-phosphate by acylation in the presence of sn-glycerol-3-phosphate acyltransferase (GPATs) and acyl-CoA. Following, a second acylation of lysophosphatidic acid in the presence of 1-acyl-*sn*-glycerol-3-phosphate acyltransferase (AGPATs) and acyl-CoA will lead to PA formation (Figure 2). PA is transformed into *sn-1*,2-diacylglycerol (DAG) by a phosphatidic acid phosphatase, releasing a phosphate group (Pi). DAG is the precursor of PCs and PEs.

Likewise, PIs and PGs are synthesized from phosphatidic acid.

PC is one of the major PLs (35 to 50%) in cell membranes and is mainly synthesized in the Golgi apparatus and mitochondria [12]. PC is found in the outer layer of a cell’s membrane. The main pathway for PC synthesis is from choline via the Kennedy pathway or the cytidine diphosphate-choline (CDP−choline) pathway (Figure 3) [13]. There is another pathway for PC biosynthesis from PE following three successive methylations.

Previous studies have shown that FAs’ nature, esterified at *sn-1* and *sn-2* positions of PCs, influences one or more synthetic pathways. The CDP−choline pathway is favored when FAs are saturated with medium chain. In the case of PUFA (DHA, AA, EPA), the PE methylation route is preferred [14].

In the CDP-choline route, choline enters the cell via facilitated diffusion where it is phosphorylated in the presence of choline kinase to form phosphocholine.

In the brain, Ross et al. in 1997 [15] suggested that the human brain has two forms of choline kinase or two sites on the same enzyme, one with low affinity and the second with high affinity for choline. The relative importance of these two sites depends on the concentration of cerebral choline.

Next, phosphocholine in the presence of choline phosphatase cytidyltransferase and cytidine triphosphate (CTP) results in the formation of CDP−choline. The last step is the combination of CDP−choline with DAG (formed from phosphatidic acid as mentioned above), in the presence of microsomal choline phosphotransferase (CPM) to form PC (Figure 3).

The biosynthesis of PC from PE takes place in hepatic microsomes. PE is gradually methylated to mono-, di-, and trimethylPE under phosphatidylethanolamine N-methyltransferase (PMETase)’s action in presence of a methyl group donor, S-adenosylmethionine [16]. PE’s methylation is associated with signal transduction and AA-rich PCs’ synthesis [17].

In rat brain, methylation activities of PL have been identified in myelin [18]. PEs’ methylation is very active in axons [16]. Another study, carried out by Tsvetnitsky in 1995, recommended that rat brain myelin contains different isoforms of phospholipid N-methyltransferase (PLMTase) [19]. Brain phosphatidylethanolamine N-methyltransferase (PEMTase) activity is influenced by n-6 PUFA levels in the diet and n-6/n-3 PUFA ratios.

Different synthetic routes have been demonstrated for PE’s biosynthesis. A synthetic route similar to that of PC has been described where ethanolamine replaces choline and PE replaces PC. This synthetic pathway happens in the endoplasmic reticulum where the PL’s polar heads can interchange [20].

Another synthetic route for PE consists of PS decarboxylation in mitochondria in the presence of PS decarboxylase (Figure 4).

PSs are formed from PEs by a calcium-dependent base exchange reaction in endoplasmic reticulum [21]. This reaction can also occur by exchanging the choline in PC with serine. Two specific enzymes are known: PS synthase-1 (PSS1) which exchanges choline and serine, and PS synthase-2 (PSS2) which replaces the ethanolamine function with serine [22].

PIs are synthesized from inositol and CDP-DG in the presence of PI-synthase following a synthetic route similar to that of PC described in Figure 3.

Concerning PGs synthesis, first, CDP-DG reacts with glycerol-3-phosphate to form PG-phosphate, which is then hydrolyzed to form PG [23]. Cardiolipids are diphosphatidyl-glycerol in which two phosphatidic acids symmetrically esterify the two primary alcohol functions of the same glycerol molecule and are formed in mitochondria from PG and CDP-DG in the presence of cardiolipin-synthase [24]. Cardiolipids, which constitute 9 to 15% of the cardiac membrane, were revealed in bovine hearts by Pangborn in 1942 and their biosynthesis was described in a rat’s heart in 1994 [25].

In addition to all GPs mentioned above, plasmalogens constitute a particular class of membrane GPs with a unique structural characteristic vinyl ether group, -0-CH=CH-, at *sn-1* position of glycerol instead of the usual ester function. FAs (C16:0, C18:0, and C18:1) are predominant at the *sn-1* position of glycerol while FAs’ composition at *sn-2* position varies with tissues.

Plasmalogens are a minority in organs such as the liver and the kidney whereas they are abundant in myocardium and brain. In neurons, plasmalogens rich in PUFA (AA and DHA) are the most abundant and ethanolamine plasmalogens (EtnPlsm) represent 40–60% of the brain PE [26].

### 2.2. Remodeling of Gylcerophospholipids

In addition to de novo synthesis, alternative pathways are responsible for remodeling FA composition of PLs. Lands and Merkel proposed a phospholipid remodeling mechanism through a de-acylation/re-acylation cycle known as the Lands cycle (Figure 5) [27]. Another PLs’ remodeling mechanism involves transacylation reactions dependent or independent of coenzyme A (Figure 6). These phospholipid-remodeling mechanisms occur in the presence of several enzymes, some of which have substrate specificities for SFA or PUFA and for various PL subclasses. In addition to Lands cycle and transacylation pathways, PL’s remodeling also involves different phospholipases by releasing PUFAs from cell membranes’ GPs.

#### 2.2.1. De-Acylation/re-Acylation Pathways

PL’s remodeling pathway, or Lands pathway, requires release of fatty acyls by a phospholipase A_1_ (PLA_1_) or A_2_ (PLA_2_), thus generating a Lysophospholipid (Lyso-PL) (Figure 5). Esterification of a new FA in Lyso-PL then takes place following two stages.

First, FA is activated as acyl-CoA. The reaction is catalyzed by an acyl-CoA synthetase and requires ATP and Mg^2+^. Acyl-CoA synthetase activity has been shown to be higher for AA in most tissues than other FA [28,29]. DHA-CoA synthetase activity has been measured in extracts from brains [30] and from rat retinas. Another enzyme, acyl-CoA hydrolase, catalyzes fatty acid and coenzyme A’s dissociation.

The second step is acyl-CoA’s transfer to a Lyso-PL. This step is catalyzed by an acyl-CoA: Lyso-PL acyltransferase. Acyl-CoA: 1-acyl-2-lysoPC acyltransferase activity has been described in cytosolic and microsomal fractions of many tissues and cells in different species [31].

**Figure 5 ijms-23-03969-f005:**
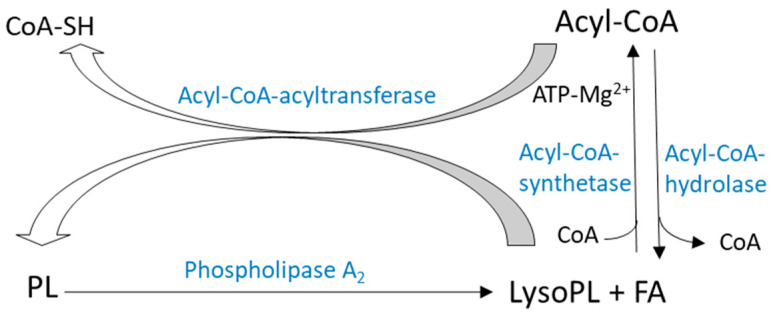
De-acylation/re-acylation cycle (Lands pathway) [31]. Lands pathway is a phospholipid remodeling mechanism through the de-acylation/re-acylation cycle where Lyso-PL is produced from PL through PLA_2_ action or from acyl-CoA through acyl-CoA-hydrolase’s action. Lyso-PL with the action of acyl-CoA-acyltransferase and in the presence of FA leads to PL production. (Copyright © 2022, Elsevier).

#### 2.2.2. Transacylation Pathways

In addition to acyl-CoA catalyzed de-acylation/re-acylation pathways described above and illustrated in this section (Figure 6A), Lyso-PL’s re-acylation can also be carried out by a transacylation mechanism using PL (Figure 6B,C).

Various enzymes called transacylases that also have substrate specificities for transferred FA and for “donor” or “acceptor” PLs catalyze this reaction.

Two transfer processes have been described, one being dependent on coenzyme A, the other not.

CoA-dependent transacylation (Figure 6B) was first demonstrated in rabbit liver microsomes and has since been found in most tissues and cell types [32]. This activity does not exhibit strong substrate specificity.

On the other hand, CoA-independent transacylation (Figure 6C), demonstrated by Kramer and Deykin in 1983, is more specific for long-chain PUFAs (AA, EPA, and DHA) [33]. In particular, it has been shown that, in the alveolar macrophages of rabbits and rats’ brain, CoA-independent transacylases selectively transfer of DHA to *sn-2* position of lyso-plasmalogens-PE [34].

Figure 6D represents lysophospholipase/transacylase activity. An acyl group at *sn-1* position of one Lyso-PL is transferred to the *sn-2* position of another Lyso-PL with the action of lysophospholipase/transacylase resulting in PL and phosphoglyceride production.

All mentioned specific remodeling activities are illustrated in Figure 6.

**Figure 6 ijms-23-03969-f006:**
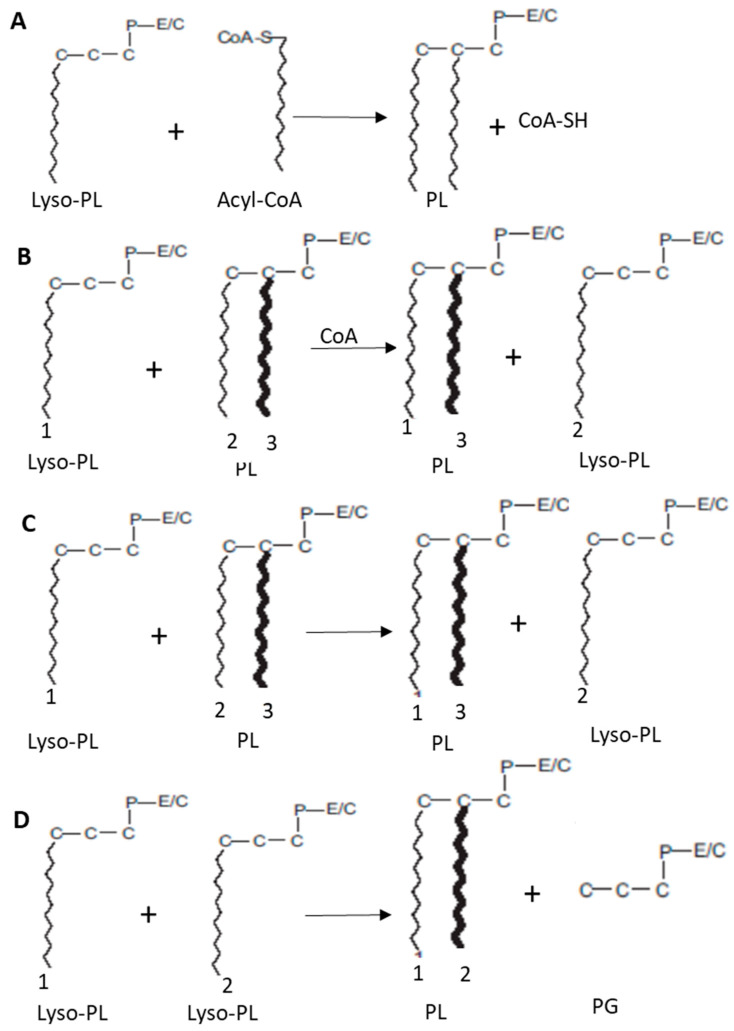
Remodeling of glycerophospholipids (E/C: ethanolamine/choline) [11]. Several pathways of remodeling activities are illustrated. (**A**): Acyl-CoA; (**B**): CoA-dependent transacylation; (**C**): CoA-independent transacylation; (**D**): Lysophospholipase transacylase activity. Reprinted with permission from Ref. [11]. Copyright 2022, Elsevier and Copyright Clearance Center. More details on “Copyright and Licensing” are available via the following link: https://www.elsevier.com/authors/permission-request-form (access on 7 January 2022).

#### 2.2.3. Remodeling Activities in the Brain

The brain possesses several enzymes responsible for FA remodeling in cerebral PLs. For example, plasmalogen (rich in AA and DHA) synthesis is carried out by the de-acylation/re-acylation route [35]. Acyl-CoA lysophospholipid acyltransferase activity has been observed in various brain entities such as microsomes and cell nuclei in the cerebral cortex of fifteen day old rabbits, the myelin in six to eight week old rats, and human brains [36,37,38]. Acyl-CoA lysophospholipid acyltransferases from microsomes and nuclei of cerebral cortical cells have different affinities for arachidonyl-CoA, suggesting the existence of two different enzymes [36].

Moreover, Vaswani and Ledeen in 1989 described, in myelin, AA and oleic acid transfer, previously activated to acyl-CoA, to Lyso-PC, and to Lyso-PI [38]. AA-CoA is more reactive than oleoyl-CoA towards Lyso-PCs and acyl-CoA: lysophospholipid acyltransferase, using oleoyl-CoA, is more important for Lyso-PC than for Lyso-PI.

Ross and Kish in 1994 worked on the human brain and showed that acyl-CoA: lysophospholipid acyltransferase has different affinities depending on the substrate and that arachidonoyl-CoA’s affinity exceeded that of palmitoyl-CoA by three times [37]. In addition, the level of acylation depended on lysophospholipids’ class according to the order Lyso-PI > Lyso-PC > Lyso-PS > Lyso-PE.

In brain tissues, CoA-dependent and CoA-independent transacylations also exist with an acyl selectivity of CoA-dependent transacylase such as acyl-CoA: lysophospholipid acyltransferase. This enzyme prefers 1-acyl-2-lysoPC over 1-acyl-2-lysoPE as acceptor PL. CoA-independent transacylase in rat brain microsomes is specific for AA and DHA [34].

These data revealed that the brain is equipped with the necessary enzymes for FA remodeling in PLs.

Figure 7 summarizes DHA’s de-acylation, re-acylation, and transacylation pathways in the brain. The cerebral DHA pool is maintained by the action of key enzymes, such as phospolipase A (PLA), lysophospholipid acyl transferase (LPLAT), glycerol-3-phosphate acyltransferase (GPAT), and 1-acylglycerol-3-phosphate-O-acyltransferase (AGPAT). These enzymes are involved in PL and LysoPL’s metabolism in the brain. In addition, the transfer of the esterified form of DHA from blood to the brain through the BBB keeps this pool constant. This transfer is facilitated by Mfsd_2a_ receptor when DHA is esterified in LysoPC-DHA. However, the transfer of other LysoPL-DHA including LysoPE-DHA and LysoPS-DHA is still not clear.

## 3. Properties and Functions of Phospholipids and Lysophospholipids with Omega-3 PUFA in the Brain

### 3.1. Brain Phospholipids and Their Fatty Acid Composition

PLs are the most abundant lipids in the brain (97 nmol/mg) compared to other organs such as the liver, the heart, and the kidney (Table 1). PE and phosphatidylethanolamine plasmalogen (PEP) represent the major classes of PL in the brain (55%), followed by PC (31%), PS (8%), and PI (5%). Traces of PG and CardioLipids (CL) have also been observed in rats’ brain [39].

PLs are considered an effective form for brain enrichment with PUFAs, especially omega-3 fatty acids. Because of their independent digestion from bile salt, PL-PUFAs are absorbed more efficiently in the small intestine, offering the best bioavailability compared to other forms of ω3-PUFA.

In the brain, PUFAs, mainly DHA and AA, are mostly found in PE species (including PEP) with PE 18:0-2:6 (11%) and PE 18:0-20:4 (8%). However, in PC, monounsaturated fatty acids (MUFA) and saturated fatty acids (SFA) are the major FAs with 54% compared to PUFAs (15%). PS contains more DHA than PC (51%) (Figure 8). In PI, AA is the major PUFA compared to DHA and EPA [39].

PE and PEP constitute the major storage form of DHA in the brain where PEP facilitates the signal transduction of bioactive mediators and can protect PUFAs, especially DHA, from oxidation in gray matter [40,41,42]. Furthermore, PC and PS are the major forms for DHA accretion to the brain.

In the brain, de novo synthesis of PL with omega-3 PUFAs can occur. As previously explained in Figure 2, in de novo synthesis of di-acyl phospholipids, the synthesis occurs in the endoplasmic reticulum and requires three major key enzymes (GPATs, AGPATs, and PAPs) as well as two precursors (*sn*-glycerol-3-phosphate and Acyl-CoA). Phosphatidic acid (PA) and lysophosphatidic acid (Lyso-PA) produced are considered as major intermediate precursors of PL biosynthesis in the brain [43,44].

Additionally, DHA carried to brain could be esterified by GPATs (*sn-1* position) and AGPATs (*sn-2* position) activities in PL during de novo biosynthesis thus suggesting that PL-DHA cerebral accretion could be partially regulated by these enzymes and PA/LPA pool in the brain [11].

However, DHA’s incorporation in brain PL through de novo synthesis is not clear and other processes such as PL remodeling could interfere in PL-DHA synthesis.

Major enzymes involved in this remolding are lysophospholipid acyltransferase (LPLAT), specifically LPCAT2/3 and LPCAT4 with Acyl-CoA specificity on EPA and DHA, respectively. This reaction is mediated by phospholipase A_2_ (PLA_2_) action and all enzymes’ activity allow the cerebral increase of omega-3 PUFA [45,46].

### 3.2. Cerebral Lysophospholipids with Omega-3 PUFA

Lysophospholipids (Lyso-PLs) are GPs in which one acyl chain is missing and one hydroxyl group of the glycerol backbone is acylated.

In 1-Lysophospholipids (1-Lyso-PLs), the acyl chain is found at position 2 (*sn-2* position), whereas in 2-lysophospholipids (2-Lyso-PLs), position 1 (*sn-1* position) is acylated.

Different forms of 1-Lyso-PLs with DHA moiety at *sn-2* position are illustrated in Figure 9. Generally, 2-Lyso-PLs are thermodynamically more stable than 1-Lyso-PLs due to the presence of a primary alcohol group at *sn-1* position.

In vivo, Lyso-PLs are produced through enzymatic hydrolysis of PLs and are involved in phospholipids’ metabolism and function as second messengers, displaying several biological functions.

#### 3.2.1. Lysophosphatidylcholine

Lyso-PCs are secreted by the liver then distributed to tissues. They are a source of PUFAs, mainly to the brain [47].

Lyso-PCs are the most abundant Lyso-PL in blood. In plasma, their concentration is closer to 1 mM [48,49]. Lyso-PCs are linked to albumin and lipoproteins in plasma and red blood cells [50]. However, their formation in plasma is not fully understood despite four different sources of plasma Lyso-PCs having been described:By hepatic lipase (LH) action which hydrolyzes PLs found in lipoproteins (high density lipoprotein (HDL)) and prefers PE to PCs. LH plays little role in Lyso-PC’s plasma production pool [51].By lecithin-cholesterol acyltransferase (LCAT) action, which is secreted by the liver and circulates in plasma mainly linked to lipoproteins, in particular high density lipoprotein (HDL) and in a minority to low density lipoprotein (LDL). LCAT catalyzes FA’s transfer at *sn-2* position of a PC found in HDL to the 3-hydroxyl group of cholesterol to form cholesterol ester and Lyso-PC. Researchers suggested that the positional specificity of human LCAT is changed when 16:0-22:6-PC concentration is improved following DHA supply [52,53]. After a diet rich in DHA, *sn-2*-lysoPC-DHA level in plasma increases 3.5 times and that of 16:0–22:6-PC increases by 12% without any modification of *sn-2*-lysoPC-16:0.Highly unsaturated Lyso-PCs are also secreted by the liver after the ingestion of unsaturated TG [54]. However, exact mechanisms of Lyso-PC secretion by liver are not well established.Chen et al., 2007 proposed a plasma synthesis of Lyso-PC due to endothelial lipase (LE) activity [55].

Most FAs at *sn-1* or *sn-2* position are PUFAs (EPA or DHA) and/or SFAs (palmitic acid or stearic acid). Lyso-PC-DHA is the Lyso-PL form mostly abundant in the brain more than other organs such as the liver, retina, heart, and kidney. Lyso-PCs have several biological effects. They act on multiple targets involved in neurodegenerative diseases, cardiovascular diseases, cancer, and neuronal apoptosis [56,57].

Moreover, both pro- and anti-inflammatory properties of Lyso-PC have been demonstrated. In vitro, pro-inflammatory activities of Lyso-PCs include the stimulation of chemotaxis in human T lymphocytes and MCP-1 (monocyte chemoattractant protein 1) cytokines production by HUVEC cells (human umbilical vein endothelial cells) [58,59], and interleukin cytokines IL-6 and IL-8 by coronary artery cells [60].

Additionally, Lyso-PCs have cytotoxic effects and can accumulate under pathological conditions such as atherosclerosis [61]. In fact, Lyso-PCs are detected at atherosclerotic lesions and are responsible for cytokine secretion, inducing inflammations of lesions. They activate the recruitment of CD^4+^ T lymphocytes in atherosclerotic lesions, according to an NF-KB cells mechanism, by increasing CXC chemokine receptor type 4 (CXCR4) receptor’s expression, thus participating in the development of atherosclerosis [62]. All these data suggest the pro-inflammatory nature of Lyso-PC.

Regarding Lyso-PCs’ anti-inflammatory activity, researchers showed that Lyso-PCs attenuate the inflammation by increasing neutrophils’ antimicrobial activity by acting on their stock of hydrogen peroxide (H_2_O_2_), reducing the interleukin 1β (IL-1β) level, and circulating tumor necrosis factor α (TNFα) [63]. Other studies have shown that Lyso-PCs reduce inflammation by acting intracellularly and by increasing the endothelial protective factors such as NOS (nitric oxide synthase) during septic shock [64].

In addition, Lyso-PC with n-3 PUFA revealed an anti-inflammatory function provoked by Lyso-PC-SFA [65]. For Lyso-PC-EPA and Lyso-PC-DHA, this effect was explained by the downregulation of monocyte action. In addition, Lyso-PLs inhibit the pro-inflammatory mediators’ (IL-5, IL-6) development stimulated by Lyso-PC16:0 and upregulate anti-inflammatory mediators (IL-4 and IL-10).

Previous studies highlighted the crucial role of Lyso-PC with n-3 PUFA in brain development and neuronal cell growth. Researchers studied the lack of Mfsd2a receptor and confirm the vital role of Lyso-PC-DHA in developing brain [66,67].

More recently, studies shows the neuroprotection role of Lyso-PC-DHA/EPA diets to increase the brain DHA and advance memory-related behavior in mice expressing human Apolipoprotein E4 (APOE4) [68,69,70]. Indeed, APOE4 human genotype was related to cognitive decline and risks of age-related neurological disorders. A decrease of APOE4 could be related to low levels of DHA, thus suggesting that a long-term treatment rich with DHA can protect from neurodegenerative disease.

Lagarde et al., 2001, by using radiolabeled DHA, showed that DHA was preferentially carried to the brain by Lyso-PC in comparison to the unesterified form of DHA. The uptake of Lyso-PC-DHA through the in vitro model of BBB is carried by one receptor highly expressed in CNS, named ‘major facilitator superfamily domain-containing protein 2A’ (Mfsd2a) [71,72,73].

In addition to the neuroprotection role, Lyso-PC-DHA was shown to be protective against retinopathy and further eye syndromes [74]. In fact, authors suggest that DHA can be proficiently improved by dietary Lyso-PC-DHA, not by TAG-DHA or non-esterified DHA. These results provide a novel nutraceutical approach for the prevention and treatment of retinopathy and neurodegenerative diseases.

Finally, Lyso-PC-DHA may be a potential treatment for depression [75]. Through a diet rich in Lyso-PC-EPA, growth of both cerebral and retinal EPA and DHA were observed. In addition, high expression of brain-derived neurotrophic factor (cyclic AMP response element binding protein) and serotonin receptor (5-hydroxy tryptamine1A) were observed in the brain. Since Lyso-PC-EPA improved EPA and DHA in the brain, it might assist in the treatment of depression.

Despite all the biological effects previously mentioned, the outcomes of clinical lipidomic studies of Lyso-PC have been controversial due to the enzymatic machinery involved in Lyso-PC metabolism [76].

#### 3.2.2. Lysophosphatidylethanolamine

Lyso-PEs, the second highest Lyso-PL after Lyso-PC with 10–50 µM in the blood, represent approximately 1% of total serum PLs [77].

Lyso-PEs are involved in cell differentiation and migration in a similar way to Lyso-PCs. Therefore, Lyso-PEs induce Ca^2+^ flux signaling by using lysophosphatidic acid receptor 1 (LPAR1) in breast cancer cells. In addition, they stimulate chemotactic migration and cellular invasion of SK-OV3 ovarian cancer cells [78,79].

Neurite growth’s simulation involves 18:1 Lyso-PE through signaling cascades in cultured cortical neurons by activating the mitogen-activated protein kinase (MAPK)/extracellular signal-regulated kinase (ERK) path [80].

Due to 18:1 Lyso-PE abundance in brain tissues, Hisano et al. demonstrated that Lyso-PEs encourage neurite outgrowth and defend against glutamate toxicity in cultured cortical neurons, thus suggesting the potential physiological function of 18:1 Lyso-PE in brain [81].

The anti-inflammatory action of 2-PUFA-1-Lyso-PE was studied in rice models after oral administration of 2-DHA-1-Lyso-PE or 2-AA-1-Lyso-PE [82]. This oral administration inhibited the plasma leakage in mice. Moreover, the anti-inflammatory interleukin’s effect with Lyso-PE-DHA was higher than with Lyso-PE-AA. The anti-inflammatory actions of 2-PUFA-1-Lyso-PE drastically reduced leukocyte infiltration suggesting that these lipids express pro-resolving activity. The authors concluded that polyunsaturated Lyso-PE may be considered effective anti-inflammatory lipids.

In the brain, Lyso-PEs are derived from PE, PE being the storage form of PL containing DHA [4]. LPE acyltransferase 2 (LPEAT2) includes DHA into PLs and is responsible for modulating DHA/EPA ratios of phospholipids [83].

However, LPEAT2 over-expression in brain cells induces cell death DHA-dependently. It was demonstrated that Lyso-PE-DHA could be a hippocampal indicator of post-ischemic cognitive impairment [84]. Indeed, in rat ischemic models, Lyso-PE had important higher levels in ischemia than a sham group. Specifically, Lyso-PE 18:1, 20:3, and 22:6 species were increased in the ischemic conditions. On the other hand, the receptor of Lyso-PE may be located in the plasma membrane to import circulating classes [83]. Lyso-PC receptor Mfsd2a is also a receptor for Lyso-PE to mediate DHA in the brain. However, the way of plasma Lyso-PE transfer to brain and transfer across the BBB is still not clear and needs further investigation.

#### 3.2.3. Lysophosphatidylserine

PS metabolism can generate Lyso-PS through phospholipase and acyltransferase activities by suppressing a FA from *sn-1* or *sn-2* position [85,86,87].

Lyso-PS is located in the brain, heart, lung, liver, kidney, and colon [88,89]. In addition, Lyso-PS is abundant in immune system organs such as the spleen, thymus, and peripheral lymph tissues. The total amount in these major organs is 1–10 µg/g. An extremely small amount is found in blood. However, Lyso-PS is higher in serum, suggesting Lyso-PS’s production during blood coagulation [90]. Lyso-PS derived from PS contains several FAs with different carbon chain length and unsaturation (16:0, 18:0, 22:6n-3, etc.). The majority of short and SFAs are esterified at *sn-1* position whereas PUFAs are esterified at *sn-2* position [90]. This distribution of FAs between *sn-1* and *sn-2* position may have a biological significance since Lyso-PS receptor activation can be related to FA position in isomers [91].

Lyso-PS metabolism involves several enzymes from the α/β-hydrolase domain (ABHD) family including ABHD16A and ABHD12 that control brain Lyso-PS pathways in neurodegenerative disease [92]. Indeed, in the brain, Lyso-PS levels are decreased by ABHD16A and increased by ABHD12 in various brain regions. These two ABHDs offer new perceptions into Lyso-PS signaling in the cerebellum, the most atrophic brain region in human polyneuropathy, hearing loss, ataxia, retinitis pigmentosa, and cataract (PHARC) subjects [93]. These enzymes can have a potential therapeutic role in the altered lipid metabolism of Lyso-PS in several diseases such as cancer, diabetes, and neurodegenerative diseases [94].

On the other hand, since Lyso-PS is found in the colon, it has a novel role in resolution of inflammation, thus reporting the emerging role of Lyso-PS in inflammatory bowel disease [95].

Lyso-PS plays a role of signaling mediator because its ability to activate signaling cascades in immunological processes, like activating macrophages during inflammation response. It was reported that PS with very long-chain saturated fatty acids (VLC-SFA-PS) strongly induce intracellular signaling through recombinant T-cell receptor ligand 2 (RTL2 receptor) [96,97]. This signaling molecule can activate cytokine secretion (TNFα, cyclic AMP, etc.) during the pro-inflammatory response induced by VLC-SFA-PS [98].The study on ABHD12 knockout mice (mutations that cause the human neurodegenerative disorder PHARC) showed that these mutations disturb the principal Lyso-PS lipase function in the mammalian brain and that an accumulation of VLC-SFA-PS occurred as well as a strong induction of cerebellar microgliosis through the neuroinflammatory response in brain [92]. Additionally, the synthetic administration of VLC-SFA-PS in TLR2 knockout mice did not induce the cerebellar microgliosis concluding that VLC-LPS induces pro-inflammatory immune response and neuroinflammation observed in aging ABHD12 knockout mice [99].

Taken altogether, this pro-inflammatory response could be decreased by DHA’s anti-inflammatory effect in neurons, suggesting a new way to explore the influence of Lyso-PS-DHA as anti-inflammatory therapy in neurodegenerative diseases [100,101].

In the brain, phosphatidylserine synthase 2 (PSS2) is important for PS-DHA synthesis and accumulation [85] whereas PLA_1_ and PLA_2_ are the potential key enzymes involved in the biosynthesis of Lyso-PS [102]. However, Lyso-PS-DHA biosynthesis is not clear. A specific pancreatic lipase family PLA_1_ was capable of producing Lyso-PS-DHA (*sn-2* position) but this lipase had affinity for other PL in vitro [87,103]. On the other hand, PLA_2_ was shown to have PS lipase activity but with no evidence of PLA_2_ specificity to PS [104].

Tsushima et al. stated that cerebral accretion of DHA when esterified at *sn-2* position of Lyso-PS is more efficient than PLs. These researchers explained the molecular pathway of Lyso-PS-DHA to reach the brain. They used fetal rat brain from mothers fed with two types of PL-DHA (PC-DHA, PS-DHA) and two others of Lyso-PL-DHA (Lyso-PC-DHA, Lyso-PS-DHA). They demonstrated that Lyso-PS-DHA was the most effective lipid carrying DHA to the brain and that Lyso-PS-DHA of the mother rat passes through the small intestinal epithelial cell monolayer by opening the tight junctions. Then, the serum DHA pool level increases and passes through the fetus’ blood–brain barrier promoting DHA increases in the fetal brain [105]. These results suggested that Lyso-PS-DHA could be an effective way to provide DHA from the mother to the fetal brain.

#### 3.2.4. Lysophosphatidic Acid

Lyso-PA is a copious bioactive lysophospholipid with multiple functions in development and pathological situations.

In CNS, Lyso-PA is derived from membrane PLs and signals through the ligand of six cognate G protein-coupled receptors (GPCRs), lysophosphatidic acid receptors (LPAR1-6) expressed in central and peripheral nervous tissues. LPAR1-6 is involved in signaling cascades including the fetal cerebral cortical growth, astrocytes development, Schwann cells proliferation, microglia activation, and in brain vasculature through the BBB.

From fetal to mature life, Lyso-PA have diverse effects during CNS development and angiogenesis [106,107].

More recently, LPAR1 was demonstrated to be involved in cerebral cortex and hippocampus development through activation of the main glutaminase isoform (GLS) [108]. In fact, neuronal intracellular calcium’s increase, a primary response to Lyso-PA exposure, suggests a modification of NMDA and AMPA glutamate receptors and Lyso-PA implications in modulation of synaptic excitatory transmission.

Lyso-PA’s involvement through its specific receptors in different central and peripheral nervous system development makes this phospholipid a versatile signaling molecule which can act as a potent mediator via LPAR1 in pathological conditions such as neuropathic pathophysiology, neurodegenerative diseases, and cancer progression [107,109,110,111,112]. LPAR1 was involved in microglial activation and brain damage after transient focal cerebral ischemia [113]. Hence, LPAR1 could be a potential therapeutic target to reduce ischemic brain damage.

Aside from LPAR1, lysophosphatidic acid acyltransferase enzyme (LPAAT), mainly LPAAT4, was highly expressed in the brain with high specificity for PUFA acyl-CoA, especially docosahexaenoyl-CoA (DHA-CoA) suggesting a potential role of LPAAT4 in incorporating DHA into brain PL.

LPAAT4 might be in charge of DHA level in neural membranes in addition to its biological role in the brain regulation [114]. Indeed, Lyso-PA-DHA was significantly associated with major depressive disorder (MDD) due to its low level found in the cerebrospinal fluid (CSF) of patients with MDD. However, it was demonstrated that Lyso-PA-DHA metabolism dysfunction is not correlated with autotaxin (ATX) activity, an enzyme that produces Lyso-PA from Lyso-PC [115].

## 4. Docosahexaenoic Acid Esterified at *sn-2* Position of Phospholipids and Lysophospholipids for Brain Benefits

### 4.1. Benefits of DHA in Healthy Conditions and Brain Disorders

The health benefits of DHA are well known from fetal to old age in healthy conditions as well as several diseases including brain disorders [116].

In healthy aging women, studies on mobility and cognitive function elucidated the potential anti-inflammatory and anti-oxidant function of DHA. DHA provided health benefits and performance improvement especially in women who practiced physical activity and have increased reactive oxygen production [117,118,119].

Moreover, DHA’s role in the pathogenesis and treatment of attention deficit hyperactivity disorder (ADHD) children and adolescents was studied. In subjects with DHA deficiency, DHA supplementation improved the clinical symptoms and cognitive performances, proposing that exogenous supply of DHA as supplementation diet could be a potential treatment for ADHD [120,121].

Few experimental works have addressed DHA potential in acute stroke treatment. Chauveau et al., 2011 highlighted the therapeutic potential of engineered brain-targeting forms of omega-3 fatty acids for acute stroke treatment. The authors used multimodal MRI to evaluate in vivo the neuroprotection of DHA and by a structured phospholipid named AceDoPC^®^(1-acetyl,2-docosahexaenoyl-glycerophosphocholine) in experimental stroke in rats undergoing intraluminal middle cerebral artery occlusion (MCAO). Results showed that neuroscores in AceDoPC^®^ rats were lower than in DHA rats and that both treatments (pooled DHA and AceDoPC groups) significantly decreased lipid peroxidation as compared to controls (pooled saline and vehicle). Additionally, MRI-based evaluation confirmed the neuroprotective effect of DHA and AceDoPC^®^ in the MCAO model [122].

According to epidemiological, preclinical, and clinical studies, DHA had neuroprotective effects in several neurodegenerative disorders including Parkinson’s disease (PD) and Alzheimer’s disease (AD) [123,124,125,126]. A dietary supplementation with DHA/EPA from fish oil (800 mg/day DHA and 290 mg/day EPA) for 6 months could reduce the inflammation and depression caused by PD [127].

On the other hand, a randomized double-blind placebo-controlled trial on 60 patients with PD was performed through patients’ ingestion of flaxseed oil (1 g of omega-3) plus vitamin E (400 IU) supplements for 3 months. The study showed that this supplementation of omega-3 and vitamin E had a satisfactory influence on the Unified PD Rating Scale (UPDRS), high-sensitivity C-reactive protein (hs-CRP), total antioxidant capacity (TAC), glutathione, and markers of insulin metabolism [128].

In vivo studies in PD mice models (6-OHDA) using fish oil supplementation confirmed the protective effect of PUFA on the loss of dopaminergic neurons and highlighted the antioxidant and anti-inflammatory properties of fish oil dietary on these PD mice model [129].

Another study on DHA supplementation in the form of PS-DHA demonstrated the protective effect of DHA on PD mice (MPTP-induced mice). Moreover, DHA used as pretreatment could inhibit apoptosis via mitochondria and MAPK pathway protecting and increasing the number of dopaminergic neurons [130]. These results suggest that a diet enriched with PL-DHA could be a potential novel therapeutic candidate for the prevention and treatment of PD.

During last decades, several randomized trials had established an association between low fish and/or low DHA intake and Alzheimer’s disease (AD) [131]. They concluded that an increase intake of omega-3 PUFA could be associated with a reduction in AD risk [132,133,134]. The first trial tested omega-3 FA (1720 mg DHA and 600 mg EPA) for six months. Researchers did not observe a significant effect on moderate AD with omega-3 supplementation, but the result did not decrease the cognitive decline. However, positive effects were observed in a small group of patients with very mild AD [135]. Moreover, several data showed that DHA and its derivative neuroprotectin D1 (NPD1) could limit Aβ peptide’s aggregation at the origin of amyloid plaque formation responsible for neurons’ degeneration [136,137].

Recently, studies focused on studying lipids’ metabolism, mainly DHA and EPA, in AD mice models and neuronal cells model. Researchers showed that PCs rich in DHA and EPA affected brain function in AD and possible mechanisms of memory and cognitive deficiency. In addition, PS-DHA and PS-EPA promoted neurite outgrowth of primary hippocampal neurons, which offers significant protection against Aβ-induced toxicity. These PLs might inhibit mitochondrial-dependent apoptotic pathway and phosphorylation in AD suggesting dietary guidance for AD prevention [138].

### 4.2. Chemical, Biochemical, and Nutritional Properties of sn-2 Position

To better understand, the significance of the *sn-2* position, we highlighted in this section the chemical, biochemical, and nutritional properties of this special position, knowing that DHA is mainly esterified at *sn-2* positions of GPs.

From a chemistry view, *sn-1* and *sn-2* mechanisms are affected by several factors including the structure of the substrate and electrophile properties of compounds. *sn-1* reaction is a unimolecular reaction involving two steps where carbocation forms, whereas *sn-2* is a bimolecular reaction with a single-step process.

From a biochemistry observation, in GPs, PUFAs and SFAs are attached to the glycerol backbone at *sn-2* and *sn-1* position, respectively, whereas a phosphate group is attached to *sn-3* position.

DHA and EPA are located almost exclusively at *sn-2* position of phosphoglycerides found in the neurons membranes [139]. Moreover, these PUFA are one of the major ω-3 LC-PUFA forms in our diet, supporting the fact that the nutritional properties of DHA and EPA are spatially important for brain development and function [140].

In physiological conditions, DHA esterified at *sn-2* position of TG, PL, or Lyso-PL acts as second messengers, displaying a wide range of biological activities. These lipids with *sn-2* DHA can improve ω3-PUFA assimilation in the liver, erythrocytes, and brain, thus improving the health benefits of 2-DHA-glycerolipids [141].

As mentioned previously, researchers showed that some Lyso-PL receptors could discriminate between 1-acyl-LPLs and 2-acyl-LPLs because of the physiological conditions, suggesting the biological role of 2-DHA-glycerolipids in different tissues.

Moreover, in the brain, PS-DHA level is higher with DHA esterified at *sn-2* position than *sn-1* position. PS-DHA is more abundant in gray matter than in white matter. The cerebral PS carries mostly 18:0 (SA) at *sn-1* position and 22:6n-3 (DHA) at *sn-2* position in all brain tissues at the exception of fond in the olfactory bulb having PS 16:0-22:6 with 16:0 (PA) at *sn-1* position at 22:6n-3 (DHA) at *sn-2* position. This special distribution of PS-DHA in the brain could be explained by the selective activity of PSS2 to esterify DHA mostly at *sn-2* position [85,142,143]. These data might elucidate the region-selectivity of PS-DHA in the brain.

From a nutritional observation, DHA’s natural sources are mostly found at *sn-2* position in both sources of TG and PC. For example, in algal oil, DHA is located at *sn-3* positions of TG. However, in fish, seal, and squid oils, DHA is found at *sn-1* and *sn-3* position mostly. In PC species, usually DHA is esterified at *sn-2* position. Only a small proportion in roe and krill oil contain DHA at *sn-1* position [140,144].

During digestion, gut pancreatic lipases hydrolyze FAs in *sn-1,3* positions which are then absorbed as FFA and metabolized independently in the lumen. However, *sn-2* MAGs are absorbed intact and enter the blood circulation to serve as the primary backbone for glycolipids synthesis in the gut and the liver, especially during extensive fat absorption. Therefore, the type of fatty acid at *sn-2* position could determine the blood lipid profile and cholesterol level.

Indeed, the type of FA at *sn-2* position is significant for overall digestibility and fat absorption. For example, in human milk, palmitic acid is largely found at *sn-2* position and this is preferred for optimum absorption of fats in infants. Infant milk formula lacking the placement of palmitic acid at *sn-2* position is thus poorly absorbed compared to human milk [145].

The nutritional effects of food may not be dependent only on saturated and unsaturated FA, but also on the position of FA, *sn-1* or *sn-2* [146].

### 4.3. Cerebral Bioavailability of DHA

As discussed previously, DHA has several beneficial effects in healthy conditions and brain disorders. An important question is related to the privileged lipid carrier of DHA to the brain. Hence, we discuss different studies related to cerebral accretion of DHA in different forms, non-esterified or esterified.

DHA esterified at *sn-2* position of PS or PC was more effective to target the brain with DHA than TG-DHA [9]. However, Lyso-PC-DHA was a privileged transporter of DHA to the brain when compared to PC-DHA and TG-DHA [147].

Indeed, researchers showed through in vitro and in vivo studies that Lyso-PC-DHA was a preferred physiological carrier of DHA to the brain when compared to non-esterified DHA [6,7]. This preference was only observed in the brain but not the liver, kidney, or heart, which even shows preference for the free form of DHA [7]. Moreover, LysoPC-DHA efficiently advances brain function in mammals [50].

However, DHA esterified at the *sn-2* position of Lyso-PC, thought to be its functional position, rapidly migrates to *sn-1* position. Consequently, researchers designed such a Lyso-PC to prevent this migration, by acetylating *sn-1* position leading to structured phospholipids named AceDoPC^®^ (1-acetyl, 2-docosahexaenoyl-glycerophosphocholine), a stabilized formula of the physiological Lyso-PC-DHA [148].

In vitro and in vivo, the cerebral accretion of AceDoPC^®^ was investigated and results showed that AceDoPC^®^ was a preferred carrier of DHA to the brain in comparison to PC-DHA and non-esterified form of DHA [8]. Despite this preference, AceDoPC was partially found in the form of Lyso-PC-DHA in the brain. Lyso-PC-DHA’s formation may be due to the hydrolysis of the acetyl moiety by the action of the endothelial lipase, a phospholipase A_1_ (PLA_1_)-like enzyme that has been shown to be expressed and secreted in brain capillary endothelial cells (BCECs) and to have a preference for DHA-containing phospholipids at the *sn-2* position [149]. Recently, a clinical study regarding the bioavailability of 13C-labeled DHA after oral intake of a single dose of 13C-AceDoPC^®^, in comparison with 13C-DHA, was conducted. Researchers showed that 13C-DHA enhancement in plasma phospholipids was considerably greater after intake of AceDoPC^®^ compared with TG-DHA. In red cells, 13C-DHA enrichment in PC was similar from both sources of DHA, with a maximum after 72 h, whereas 13C-DHA enrichment in PE was greater from AceDoPC^®^ compared to TG-DHA, and increase after 144 h. This study indicated that DHA from AceDoPC^®^ is more efficient than from TG-DHA for a continuous accretion in red cell PE and brain [4].

Taking all these works together, we can conclude that different mechanisms could explain the brain bioavailability differences between non-esterified form of DHA, Lyso-PC-DHA, AceDoPC^®^, PC-DHA, PS-DHA-PS, PE-DHA, and TG-DHA.

On the other hand, long-chain PUFAs’ transport in preterm infant plasma was shown to be dominated by PC [150]. Recently, an in vivo study on rats determined the way EPA impacted brain DHA levels [151]. Using 13C-EPA, researchers found that newly synthesized 13C-DHA was mainly converted and stored as PE-DHA (37–56%) in the liver, secreted in plasma as TG-DHA, and accumulated in the brain as PE-DHA and PS-DHA, proposing a significant role for EPA in maintaining the brain DHA levels.

## 5. Discussions

Neurodegenerative diseases including Alzheimer’s and Parkinson’s diseases are associated with high morbidity and mortality rates. Currently, there is no treatment that can cure these diseases although many drugs have been produced to slow their development.

A number of researchers have highlighted the beneficial effect of omega-3 PUFAs, particularly DHA, through their involvement in several biochemical functions in the brain.

In fact, almost 60% of PUFAs in neuronal membranes comprise DHA, thus representing the most common PUFA in the human brain [126]. DHA in nerve cell membranes is captured by the brain through blood circulation, either synthesized by the liver, or brought directly from the diet [2]. Several pathways are involved in the structural distribution of DHA in the brain from DHA’s de-acylation, re-acylation, and transacylation pathways as explained in Figure 7.

DHA is possibly the utmost pleiotropic molecule in nerve cell membranes since it is implicated in numerous essential functions ranging from structural components to brain homeostasis and regulation of neurogenesis until the indirect antioxidant function of DHA.

Concerning the structural and physicochemical properties of DHA in nerve cell membranes, as previously discussed in Section 3.1 on brain phospholipids and their fatty acid composition, DHA is esterified at *sn-2* position of PE and PEP, the most abundant phospholipid in nerve cells. This molecular combination is significant to ensure the structural and physicochemical properties of cell membranes. In fact, DHA modulated the microviscosity, passive permeability, lateral mobility, lipid–protein and protein–protein interactions, and conformational transitions of membrane proteins [152].

Moreover, DHA is crucial for brain homeostasis during fetal development as well as throughout the lifespan as a neuromodulator of nerve cell function through the modulation of differentiation of novel neurites and synapses, refinement of synaptic connectivity, neurotransmitter release, and memory consolidation processes [153].

Additionally, DHA has been identified as an indirect antioxidant through modulation of gene expression within thioredoxin and glutathione antioxidant systems. Deregulation of these self-protecting defenses managed by DHA is associated with the development of neurodegenerative diseases [152].

Diaz et al., 2021 suggested a possible hypothesis that under oxidative conditions, a fraction of DHA in the brain parenchyma may undergo a non-enzymatic oxidation to produce a specific DHA-derived peroxyl radical 4-hydroxy-2-hexenal (HHE). HHE activates a nuclear factor erythroid-related factor 2 (Nrf2), a key regulator of transcriptional activation of antioxidants in the brain [152].

To better understand the potential therapeutic influence of DHA transported by different carrier forms to the brain, we first summarized the wide range of GPs found in cell membrane as well as their biosynthesis and remodeling focusing on de-acylation, re-acylation, and transacylation pathways.

Next, we highlighted different properties of PLs and Lyso-PLs carrying DHA to the brain, knowing that the brain has several enzymes responsible for remodeling FA composition of cerebral PLs.

Indeed, the brain contains the highest content of PLs in comparison to other organs. Among these PLs, PE-DHAs are the most abundant, thus constituting the major storage of DHA although PC and PS are also known to transport DHA to the brain.

We discussed the mechanism of formation and physiological role of diverse Lyso-PL-DHA including Lyso-PC, Lyso-PE, Lyso-PS, and Lyso-PA. Among these, Lyso-PCs are the most abundant Lyso-PL in blood. They have several biological effects mainly pro- and anti-inflammatory properties and are crucial for brain development and neuronal cell growth when DHA is esterified in the glycerol backbone. In addition, Lyso-PC-DHA showed a neuroprotective effect and a protection against retinopathy and other eye diseases. Moreover, Lyso-PC-DHA might be a potential treatment for depression.

Another bioactive form of Lyso-PL is Lyso-PE constituting the second highest Lyso-PL after Lyso-PC in blood. Lyso-PE is involved in the stimulation of neurite growth in the brain and has a potential physiological role in the brain. When DHA is esterified in Lyso-PE, these lipids are considered as a potent anti-inflammatory and could be a hippocampal indicator of post-ischemic cognitive impairment. Moreover, researchers suggested that LPAAT4 could have a potential role in incorporating DHA in the form of Lyso-PA into the brain.

As mentioned previously, several studies have discussed the cerebral accretion of non-esterified DHA and DHA esterified in Lyso-PC-DHA, AceDoPC^®^, PC-DHA, PS-DHA, PE-DHA, and TG-DHA. AceDoPC^®^ was the privileged carrier of DHA to the brain when comparing, in vitro and in vivo, the cerebral accretion between AceDoPC^®^, PC-DHA, and DHA and ex-vivo when comparing the bioavailability of AceDoPC^®^ and TG-DHA [4,8]. Moreover, it has been demonstrated that the cerebral accretion of DHA when esterified at *sn-2* position of Lyso-PS is more efficient than PC-DHA, PS-DHA, and LysoPC-DHA.

In vitro and in vivo, Lyso-PC-DHA was a preferred physiological carrier of DHA to the brain when compared to non-esterified DHA [6,7].

Although many studies have highlighted the cerebral accretion of Lyso-PC-DHA, cerebral bioavailability of other forms of *sn-2* Lyso-PL-DHA including Lyso-PE, Lyso-PS, Lyso-PI, Lyso-PG, and Lyso-PA are not well explored. Hence, several Lyso-PLs as carrier of DHA to the brain should be developed and cerebral bioavailability of these Lyso-PLs in comparison to non-esterified form of DHA can be studied in vivo as well as in vitro.

The main challenge will be maintaining DHA at *sn-2* position in synthesized Lyso-PLs and avoiding DHA’s migration from *sn-2* to *sn-1* position. A number of approaches have been defined to reduce the occurrence of acyl migration during de-acylation. These include the use of a packed bed reactor or the addition of a borax buffer in the reaction medium. The reaction temperature may also be an important parameter to check. For instance, for DHA-rich LPL synthesis, lowering the temperature from 40 °C to 30 °C during the reaction reduces DHA migration by 37% [154].

The choice of appropriate lipase when hydrolyzing *sn-1* position of PLs to produce Lyso-PL target is another challenge. Endothelial lipase (LE) can be used since LE is a PLA_1_ of endothelial cells of BBB responsible for LysoPL and DHA’s generation in plasma. LE showed specificity towards polar head of PL: PE > PC > PS and a preference for species containing DHA at *sn-2* position [55]. During the Lyso-PL-DHA synthesis process, thin layer chromatography (TLC) and analytical reverse phase high performance liquid chromatography (HPLC) can follow the reaction.

Concerning DHA transport to the brain across the BBB, different mechanisms are known in the literature, taking into consideration that BBB is a physical and metabolic barrier with high selectivity and specificity regarding the passage of molecules across it to the brain [2]. Figure 10 elucidates several suggestions of DHA transfer in PLs and Lyso-PLs form.

Generally, in plasma, DHA is linked to proteins including albumin and lipoproteins. A dissociation of albumin and lipids occurs to free the lipid content and allow DHA transfer across BBB whereas the combination of lipoproteins and lipids can pass across endothelial cells through lipoprotein receptors and result in a facilitated transport process (Figure 10). Other facilitated transport processes through the BBB involve cerebral membrane proteins such as fatty acid transport protein (FATP) expressed in various parts of brain and CD36 expressed in endothelial cells [2].

Concerning PL-DHAs passage across the BBB, a simple passive diffusion of these molecules through the phospholipids bilayer of endothelial cells can occur. Then, through the action of phospholipases, free fatty acids, and Lyso-PLs are generated. In addition, PL-DHAs can pass between two adjacent endothelial cells through tight junctions: occludin, JAM, cadherin, and claudin. Reversible “flip-flop” movements are also well known.

A recent study highlighted the role of major facilitator superfamily domain-containing protein 2 (Mfsd2a), expressed in BBB endothelium, as a major transporter of Lyso-PC-DHA to the brain. In fact, Mfsd2a is a membrane transport protein expressed in the endothelium of BBB and has an important role in BBB formation and function. Importantly, Mfsd2a transports DHA into the brain only in *sn-2*-LysoPC-DHA form. However, it was reported that *sn-1*-DHA-Lyso-PC is also efficient in carrying DHA to the brain, suggesting that brain receptor Mfsd2a is not specific only to *sn-2*-DHA form but also to *sn-1*-DHA form [149].

Following these results, if Mfsd2a was able to transport Lyso-PC-DHA, will the same transporter carry different types of Lyso-PL-DHA to the brain? Answering this question will clarify whether the choline moiety at *sn-3* position is critical for the passage of DHA across the BBB or not.

Finally, production of DHA-rich LysoPLs and study of their passage mechanism across BBB could be of great relevance to pharmaceutical applications since most commercially available DHA formulations are in the form of PLs or TGs, which are not the privileged carriers of DHA to the brain. These molecules can lead to new therapeutic targets for neurodegenerative diseases.

## 6. Conclusions

In the context of prospective prevention and treatment of neurodegenerative syndromes, research on DHA benefits as well as transport to the brain has grown significantly in the last years. However, mechanisms on the bioavailability and DHA incorporation into the brain should be more explored. Although many studies have suggested that DHA is transported to brain by DHA transporters more specifically than others, future studies are required to explore a precise carrier molecule for improving brain DHA and this specific carrier might be used as a substitute in the prevention and treatment of neurodegenerative disorders.

## Figures and Tables

**Figure 1 ijms-23-03969-f001:**
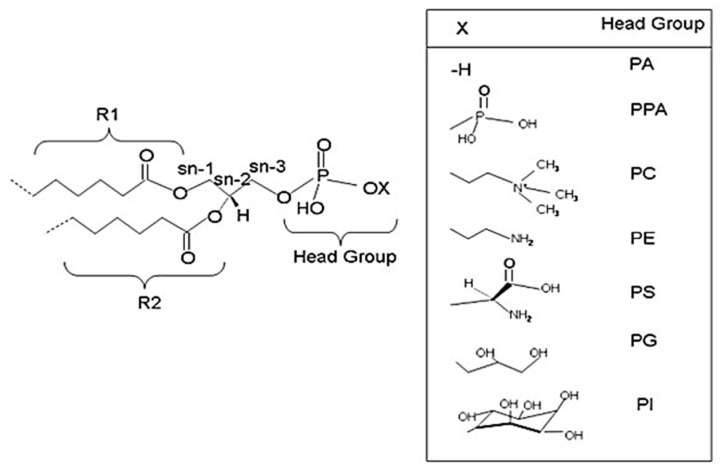
Structures of glycerophospholipids (GPs). R1, R2, and X are variables at *sn-1*, *sn-2*, and *sn-3* positions, respectively. Depending on the head group at *sn-3* position, several GPs are formed. PA: phosphate; PPA: pyrophosphate; PE: phosphatidylethanolamine; PC: phosphatidylcholine; PS: phosphatidylserine; PG: phosphoglycerol; PI: phosphoinositol [10]. (Copyright © 2022, Yetukuri et al.; licensee BioMed Central Ltd., London, UK).

**Figure 2 ijms-23-03969-f002:**
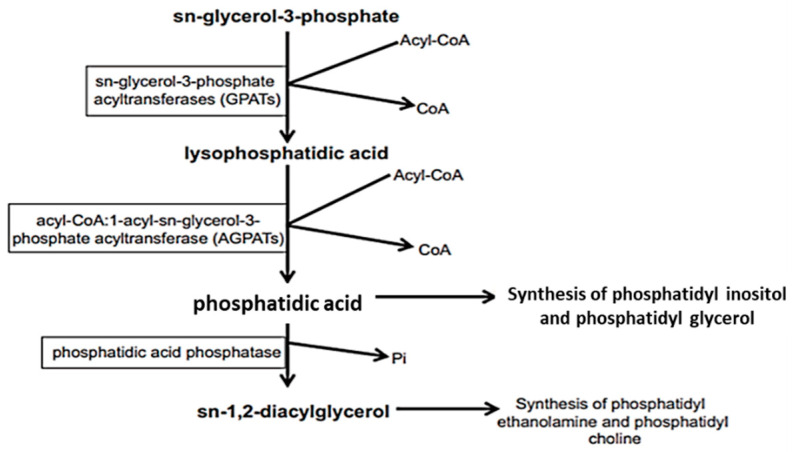
De novo synthesis of *sn*-1,2-diacyl glycerophospholipids [11]. From *sn*-glycerol-3-phosphate to the formation of *sn*-1,2-diacylglycerol (DAG), several enzymes are involved: GPATs, AGPATs, and phosphatidic acid phosphatase. DAG is the precursor of PE and PC whereas PI and PG are produced through phosphatidic acid. Reprinted with permission from Ref. [11]. Copyright 2022, Elsevier and Copyright Clearance Center. More details on “Copyright and Licensing” are available via the following link: https://www.elsevier.com/authors/permission-request-form (access on 7 January 2022).

**Figure 3 ijms-23-03969-f003:**
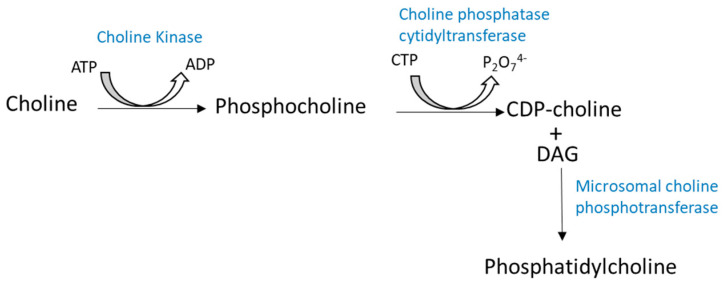
Cytidine diphosphate-choline (CDP−choline) pathway for PC synthesis. Phosphatidylcholine or phosphocholine (PC) is formed from choline in the presence of choline kinase and ATP. Then, PC leads to the formation of CDP−choline with the action of choline phosphatase cytidyltransferase. The last step concerns the combination of both CDP−choline and diacylglycerides (DG) to form PC.

**Figure 4 ijms-23-03969-f004:**
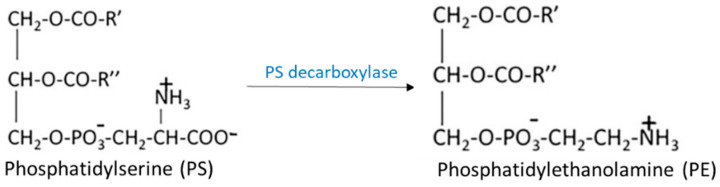
Synthetic route of PE by decarboxylation of PS. PS with the action of PS decarboxylase leads to PE synthesis in mitochondria.

**Figure 7 ijms-23-03969-f007:**
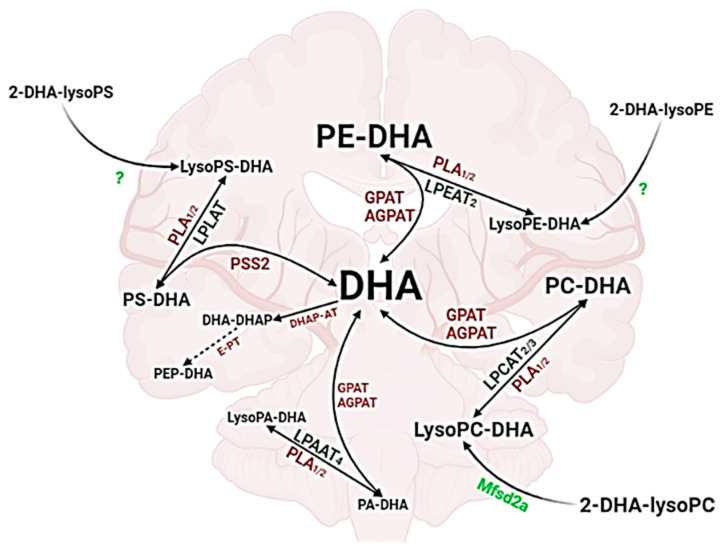
DHA’s de-acylation, re-acylation, and transacylation pathways in brain. DHA: docosahexaenoic acid; PLA: phospholipase A; LPLAT: lysophospholipid acyl transferase; LPEAT: lysophosphatidylethanolamine acyl transferase; LPCAT: lysophosphatidylcholine acyltransferase; LPAAT: lysophosphatidic acid acyltransferase; PSS_2_: phosphatidylserine synthase 2; GPAT: glycerol-3-phosphate acyltransferase; AGPAT: 1-acylglycerol-3-phosphate-O-acyltransferase; DHA-DHAP: dihydroxyacetone phosphate carrying DHA; E-PT: ethanolamine phosphotransferase (Created with BioRender.com accessed on 7 January 2022).

**Figure 8 ijms-23-03969-f008:**
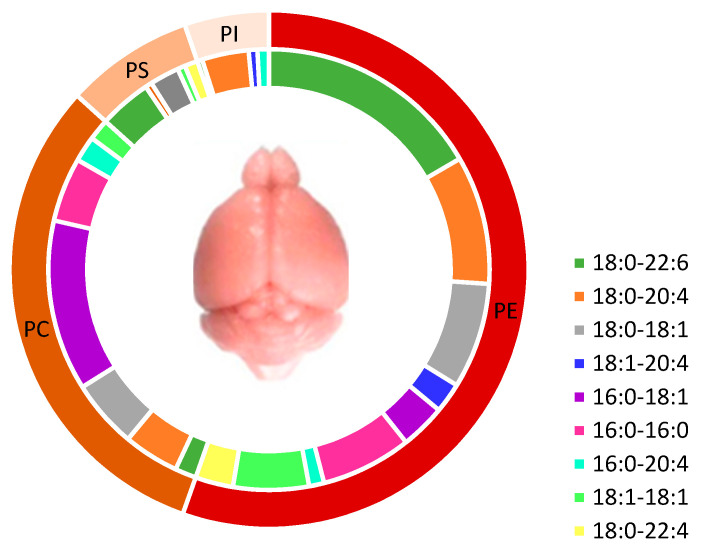
Major species of phospholipids in rats’ brains. Phospholipids content in fatty acids showed that PE has the highest content in PUFA (22:6n-3 and 20:4n-6) in comparison to PS and PC. Adapted with permission from Ref [39]. Copyright 2021, Springer Nature and Copyright Clearance Center. More details on “Copyright and Licensing” are available via the following link: https://support.springernature.com (access on 7 January 2022).

**Figure 9 ijms-23-03969-f009:**
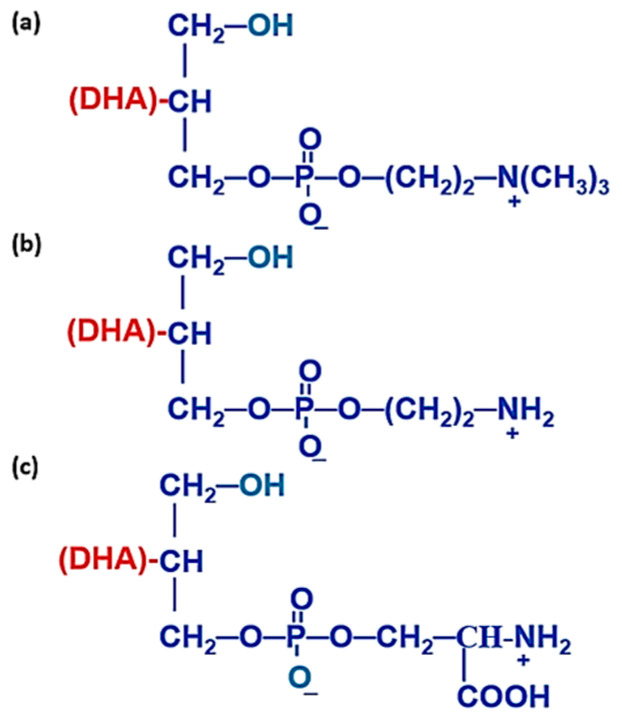
Chemical representation of some lysophospholipids with DHA moiety at *sn-2* position. (**a**) 1-lyso,2-docosahexaenoyl, glycerophosphocholine (Lyso-PC-DHA), (**b**) 1-lyso,2-docosahexaenoyl,glycerophosphatidylethanolamine (Lyso-PE-DHA), (**c**) 1-lyso,2-docosahexaenoyl, glycerophosphatidylserine (Lyso-PS-DHA).

**Figure 10 ijms-23-03969-f010:**
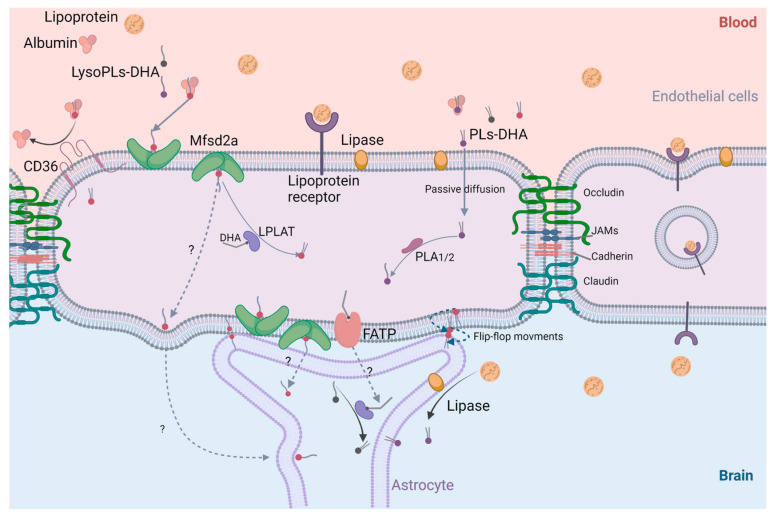
Mechanisms of PLs-DHA and Lyso-PLs-DHA passage across BBB. Mfsd2a: major facilitator superfamily domain-containing protein 2; CD 36: cluster of differentiation 36; FATP: fatty acid transport protein; PLA: phospholipase A; LPAT: lysophospholipase acyl transferase; DHA: docosahexaenoic acid; LysoPLs-DHA: lysophospholipids carrying DHA; PLs-DHA: phospholipids carrying DHA; JAM: junctional adhesion molecule (Created with BioRender.com accessed on 7 January 2022).

**Table 1 ijms-23-03969-t001:** Concentration of different phospholipids in the brain, heart, kidney, and liver of rats (nmol/mg, *n* = 4). Reprinted with permission from Ref. [39]. Copyright 2021, Springer Nature and Copyright Clearance Center. More details on “Copyright and Licensing” are available via the following link: https://support.springernature.com (access on 7 January 2022).

	PE	PC	PS	PI	PG	CL	Total
Brain	53.9 ± 5.3	30.5 ± 4.5	7.78 ± 0.93	5.13 ± 0.76	0.02 ± 0.01	0.27 ± 0.15	97.6 ± 21.6
Heat	22.0 ± 6.0	17.0 ± 5.0	0.48 ± 0.14	2.74 ± 0.86	0.17 ± 0.06	2.31 ± 0.66	44.7 ± 12.6
Kidney	24.7 ± 5.5	18.0 ± 4.2	2.31 ± 0.45	5.31 ± 1.3	0.02 ± 0.01	1.38 ± 0.35	51.7 ± 11.3
Liver	22.8 ± 3.3	31.2 ± 3.8	1.21 ± 0.32	8.52 ± 1.1	0.01 ± 0.00	1.08 ± 0.21	64.9 ± 8.6

## Data Availability

Not applicable.

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
