# Peer review of "Emerging Role of Phospholipids and Lysophospholipids for Improving Brain Docosahexaenoic Acid as Potential Preventive and Therapeutic Strategies for Neurological Diseases"

_ijms, 2022, doi:10.3390/ijms23073969_

Round 1

Reviewer 1 Report

The article " Emerging Role of Phospholipids and Lysophospholipids for  improving brain Docosahexaenoic Acid as potential preventive  and therapeutic strategies for neurological diseases" tries to review the role of DHA is neurological disorders.

This is an interesting field with much activity over the last 10-15 years and there is a number of other reviews covering the same or similar areas (e.g., DOI: doi.org/10.1016/j.plefa.2017.08.001, 10.1096/fj.201801412R,  10.3390/ijms2221118260. Although a new review with a new perspective on a field in rapid development would be welcome in my opinion this manuscript does not bring a new perspective on the subject.

A few more specific comments:

  • The manuscript covers very basis/fundamental information that in many cases  is too  obvious to be included
  • Chemical names of compounds are incorrect, e.g. instead of 1-lyso,2-docosahexaenoyl,glycerophosphocholine should be 1-lyso-2-docosahexaenoyl-sn-glycero-3-phosphocholine.
  • Besides, in structures of LPLs or PLs O and N should be charged. In general, the quality of chemical structures and most Figures is insufficient, including their resolution
  • The manuscript was prepared very carelessly with a lot of  grammatical errors, and typos, e.g. 

2.2. Remodeling of Gylerophospholipids

  • The authors write that

'Other facilitated transport process through BBB involve cerebral membrane proteins such as  Fatty Acid Transport Protein (FATP) expressed in various parts of brain, CD36 and GPCR expressed in endothelial cells'.

How GPCRs affect transport?

Indeed, DHA as a free fatty acid (FFA4) or in lysophosphatitylcholines can acts as GPCR agonist  (GPR119, GPR55) and this issue should be expanded.

In summary, I have unfortunately ended up rather negative with respect to this manuscript. I can only recommend that the authors direct the focus to the parts where they have the strongest expertise and add fragments that are not present in the previous reviews.

Reviewer 2 Report

MS# Emerging Role of Phospholipids and Lysophospholipids for improving brain Docosahexaenoic Acid as potential preventive and therapeutic strategies for neurological diseases

The manuscript by reviews the potential of DHA containing Phospholipids and Lysophospholipids for improving brain DHA uptake, and their potential as preventive and therapeutic strategies for neurological diseases. The review is well structured and elegantly written, and integrates the present knowledge on the intrinsic differences between different PL and LPO as DHA carrier’s for the high demand of nerve cells. I did enjoy very much reviewing this article and  have only few comments, some of which could help to improve the readability of the review

  • The descriptive biochemistry of DHA containing PL and LPO at the beginning of the review is well presented and very clear, which is fine for the general audience. However, in line wit this I would suggest a brief introductory paragraph outlining the essential and pleiotropic roles of DHA in nerve cell membranes, ranging from structural roles to indirect antioxidant (which is paradoxical considering it is a highly susceptible for oxidation in the brain parenchyma). This is the answer to the question “Why is it essential to optimize DHA supply?”
  • A subsection devoted to PUFA and LCPUFA transport across the BBB. The complexity and selectivity of the barrier deserves special consideration. This would be very much appreciated for researchers on nutritional requirements of the brain, molecular neurobiologists…
  • Section 4.1 can be substantially improved. For instance, the second paragraph “in old women …” appears to be out of context but is the only mention to healthy aging

Minor:

Unify 13C or 13(superscript)C lines 684-

Are you sure this is possible?  “The anti-inflammatory action of 2-PUFA-1-Lyso-PE was studied in rice models after oral administration…” Line 432 :)

In summary, this is an excellent review and I wish to congratulate the authors for the effort to summarize and update all this relevant information

Reviewer 3 Report

The review is interesting, original. The paper is well written. the conclusions consistent with the evidence and arguments
presented. 

Round 2

Reviewer 1 Report

Although the authors responded to individual points, I still believe that they misinterpreted the issues of GPCRs involvement in transport (Section 3.2.3 and Figure 10 suggesting the involvement of GPCRs). So, if the authors know that GPCRs participate in the transport of Lyso-PLs, please describe it.

Question: How GPCRs affect transport?

Response: Indeed, GPCR have been demonstrated recently to have a role as specific transporter of LysoPS-DHA. We dissuced in our review the GPCR transporter family and how it is affecting DHA transport in section 3.2.3: Lysophosphatidylserine.

Section 3.2.3 list the receptors that are targets for Lyso-PS, but do not indicate their participation in transport

Also, since the authors list GPCRs activated by Lyso-PS and Lyso-PA why they do not list such receptors for Lyso-PC (e.g., GPR119, GPR55, FFA1).

Round 3

Reviewer 1 Report

I still insist that there is no evidence for the direct transport of Lyso-PLs by GPCRs. Even in the cited article such information goes not exists (https://www.frontiersin.org/articles/10.3389/fncel.2020.00139/full).

Also, in the following paragraph the word "transport" should be replaced by e.g. can be activated  

Moreover, recent studies suggested that GPCRs can transport Lyso-PC with short and medium chain saturated fatty acids (12:0, 14:0, 16:0, 18:0 and 18:1) [Liu P et al. 2020; Drzazga et al., 2018]. 
